# The EuropaBON Stakeholder Dashboard: A dynamic web application to map Europe's biodiversity community

Christian Langer[1,2]*, Jessica Junker[1,2,3,4], Marek Giergiczny[1,2], Ian McCallum[5], Ivelina Georgieva[5], Henrique Miguel Pereira[1,2,6]

1 Institute of Biology, Martin Luther University Halle-Wittenberg, Halle, Germany, 2 German Centre for Integrative Biodiversity Research (iDiv) Halle-Jena-Leipzig, Leipzig, Germany, 3 Re:wild, Austin, Texas, United States of America, 4 Senckenberg Museum for Natural History Görlitz, Görlitz, Germany, 5 International Institute for Applied Systems Analysis (IIASA), Laxenburg, Austria, 6 BIOPOLIS Program in Genomics, Biodiversity and Land Planning, CIBIO, Campus de Vairão, Vairão, Portugal

* christian.langer@idiv.de

## Abstract

Europe's biodiversity faces increasing pressure from climate change, pollution, and habitat loss, while governments struggle to sustain the monitoring efforts required to respond effectively to these challenges. Addressing this gap calls for a coordinated and inclusive approach that brings together all relevant biodiversity stakeholders to co-design a robust European biodiversity monitoring system. To support this, the Europa Biodiversity Observation Network (EuropaBON) has established one of the most comprehensive biodiversity stakeholder networks in Europe. To analyse this community and support evidence-based improvements, we developed the Europa-BON Stakeholder Dashboard – a dynamic, interactive web application that maps and visualises the EuropaBON stakeholder network's structure in real-time. Accessible at https://europabon.org/dashboard, the dashboard enables users to explore stakeholder connections across three key dimensions: occupational sector, realm (terrestrial, freshwater, marine), and geographic region. It displays detailed network graphs, an interactive map, and summary statistics that highlight institutional positions in biodiversity data flows (e.g., data users, data providers, or both), levels of participation in EuropaBON activities, and connections to key EU projects and infrastructures. Users can identify the most central and active institutions in the network, filter and download data, and assess coverage across different thematic areas and regions. This tool supports both researchers and policymakers by offering an up-to-date overview of who is involved in biodiversity monitoring across Europe, where collaborations exist, and where further engagement is needed. By combining technological integration with stakeholder participation, the EuropaBON Stakeholder Dashboard enhances transparency, promotes inclusivity, and contributes to a more coordinated and effective biodiversity monitoring landscape in Europe.

**Data availability statement:** The entire source code of the website is available in the EuropaBON GitHub repository (https://github.com/EuropaBON/stakeholder-dashboard), and an archived version of the code has been made publicly available on Zenodo (https://doi.org/10.5281/zenodo.10047342).

**Funding:** This is a product of the EuropaBON project funded from the European Union's Horizon 2020 research and innovation programme under grant agreement No 101003553. The funders had no role in study design, data collection and analysis, decision to publish, or preparation of the manuscript.

**Competing interests:** The authors have declared that no competing interests exist.

## Introduction

In this paper, the term EuropaBON refers, depending on context, to (a) the EuropaBON project and its associated aims and activities; (b) the integrated biodiversity monitoring system being developed by the project for Europe; or (c) the network of stakeholders and members engaged throughout the project. Where relevant, we specify 'project', 'system', or 'network' to distinguish between these closely related but distinct components. This distinction is important in light of the broader context in which EuropaBON operates. One of the main challenges facing global biodiversity research lies in the scarcity and uneven distribution or fragmentation of unbiased data across both temporal and spatial dimensions and taxa [1]. European biodiversity data, while plentiful, are dispersed among various organisations, projects and individuals, hindering a comprehensive and coherent understanding of ecological patterns. The necessity for robust and representative biodiversity data is underscored by the imperative to effectively implement and assess policies aimed at conserving biodiversity and ecosystems [2]. To address this challenge, a collaborative approach is essential. By uniting biodiversity stakeholders, the Europa Biodiversity Observation Network (henceforth EuropaBON) project aimed to collectively design a monitoring framework that not only integrated existing data but also addressed the gaps in coverage [3].

This joint effort has sought to build a unified and coherent biodiversity monitoring system for improving both the reliability and accessibility of biodiversity data – such as species occurrences, habitat condition, and population trends – ultimately enabling more informed conservation strategies and evidence-based policy decisions [4]. Several global and regional initiatives have emerged in recent years to coordinate biodiversity monitoring and foster data standardisation, with the Group on Earth Observations Biodiversity Observation Network (GEO BON) playing a central role at the global level. GEO BON provides a strategic framework for the development of Biodiversity Observation Networks (BONs) worldwide and promotes the use of Essential Biodiversity Variables (EBVs) to enhance data interoperability and policy relevance [5]. EBVs are designed to monitor biodiversity status and trends across multiple spatial and temporal scales. They serve as an intermediate layer between raw biodiversity data and policy indicators, facilitating improved access to policy-relevant information [6]. Regional and national Biodiversity Observation Networks (i.e., BONs), such as the Asia BON [7], the Americas BON [8], and national efforts like SANBI in South Africa [9], contribute to this global vision. The EuropaBON network serves as the European regional BON under GEO BON, aligning its monitoring design and stakeholder engagement strategy with these broader objectives. With nearly 1,600 registered members from 704 organisations across 72 countries (as of 13/09/2024), the EuropaBON network is one of the largest and most influential biodiversity communities in Europe. Although not yet implemented, network members (henceforth "members") have been involved in every step of making recommendations- and proposing a design for the new European biodiversity monitoring system, from identifying user and policy needs, assessing existing European monitoring systems and identifying data gaps, to defining the Essential

Biodiversity Variables (EBVs) to be monitored within the system. This new system is designed in such a way as to address multiple EU policies and reporting needs [10]. Assessing the impact of key contributors, measured through their level of connectedness within the network, their participation in EuropaBON project activities (e.g., workshops, surveys, consultations), and their involvement in shaping the structure of the monitoring framework, is crucial for understanding potential geographical or thematic biases embedded in the collaboratively designed system. These forms of impact reflect both influence on decision-making processes and the degree to which institutions are contributing knowledge, capacity, or data. Recognizing such biases is essential for creating a representative monitoring system that reflects the diversity of stakeholder perspectives and ensures broad relevance and uptake across Europe. To enhance transparency, stakeholder engagement, and adaptability, we developed an interactive web-based dashboard (https://europabon.org/dashboard/) that allows users to explore the European biodiversity community, its key actors, and their connections. The dashboard visualises stakeholder relationships across three key dimensions: occupational sector (academia, NGO, governmental organisation, private sector, citizen science, other), ecological realm (terrestrial, freshwater, marine), and EU region as defined by the United Nations geoscheme.

The main objective of this study was to map the EuropaBON stakeholder network across these dimensions and identify gaps in geographic coverage, thematic focus, and areas of expertise. While previous efforts – such as the EU BON project – conducted limited and largely static analyses of institutional relationships [11], the EuropaBON project has advanced this work by offering a dynamic, continuously updated real-time tool for exploring network structure and engagement, drawing on the FAIR principles [12]. Unlike static, snapshot-based data sets that require manual updates and only provide a fixed view of data that can quickly become outdated, the dashboard ensures a continuous flow of data, automatic updates and interactive visualisations. Users are no longer limited to passively viewing the data, but can actively explore relationships, filter information and analyse trends in real time. This added layer of transparency and continuous self-assessment positions EuropaBON as a leading initiative that has contributed to building an inclusive, balanced and evolving biodiversity monitoring community.

## Methods

### Input data

The information provided by registered stakeholders (also referred to as members throughout the text) in the EuropaBON members portal [13], has served as the primary data source for developing the dashboard (https://europabon.org/dashboard) and conducting the network analysis. Membership grew rapidly in the first year following the network's establishment, partly due to three stakeholder conferences held in 2021, and continued to increase, albeit at a slower rate, throughout 2022 and 2023 (S1 Fig). Membership recruitment primarily took place through EuropaBON events, as participation was limited to registered members. Membership grew organically through event announcements and social media outreach. In addition, the project was widely recognized across Europe, leading to a high number of registrations from a broad range of stakeholders. While the database includes many key biodiversity actors, we acknowledge that some notable organisations, such as the Natural History Museum in London, are not represented. To ensure network sustainability, the members' portal and the associated dashboard were designed to be maintained for at least five years beyond the project's duration (until 2029). Members provided essential information via an online registration form (https://europabon.org/members/register/index), with the full list of questions asked during the registration process available in S1 File. Members are able to update their information at any time by logging into the EuropaBON members portal [13]. Any change in the portal, such as the registration of a new member or the modification of an existing user profile (e.g., affiliation, data provision or data usage) is reflected in the dashboard. The main component in the dashboard, the network graph, is based on institutions (henceforth referred to as "nodes") and their data exchange interactions (henceforth referred to as "edges"). This study was conducted in accordance with the EuropaBON Data Privacy and Use Policy, available from the EuropaBON website, which outlines the terms and conditions governing the collection, access, and use of stakeholder data.

## Data processing

Following a clean-up of the registration form's input fields – examples of which are provided in S2 File – we grouped individual member entries by their affiliated institution (see S3 File) to ensure consistent institutional representation on the dashboard. If an institution has sub-institutions, these are not explicitly grouped together in the data. For example, GBIF and GBIF Spain are listed as separate entities. This distinction is maintained for two reasons: first, their differing geographical locations – GBIF is based in Denmark, while GBIF Spain operates under CSIC in Spain; and second, the dashboard reflects the information exactly as provided by individual members during registration. Modifying these entries could have distorted the actual output. Individual members' information is not disclosed in the dashboard. To provide the data for the dashboard, we structured the grouped table into a JSON-based API (Application Programming Interface). JSON (JavaScript Object Notation) is an open standard file format that uses human-readable text to store and transmit data objects consisting of name-value pairs (e.g., "id": "1", where the name is "id" and the value is "1"). APIs facilitate communication via designated API endpoints, which are specific URLs for sending requests and receiving responses. The API endpoints are available in the S4 File. The API endpoint https://europabon.org/dashboard/api/nodes summarises the nodes and also contains information about the connections between the individual nodes, represented by the edges in the network graph. Each node is assigned an ID and corresponding attributes (e.g., label, country, scope, group, etc.). The S5 File shows an example of the attributes for ID 1. The attributes contain information about occupational sector (academia, NGO, governmental organisation, private sector, citizen science, other), realm (terrestrial, freshwater, marine, cross-realm) or geographic region (Northern Europe, Western Europe, Eastern Europe, Southern Europe, and non-European regions). If a node is connected to several realms (e.g., Terrestrial, Freshwater), it is referred to as "cross-realm". The nodes also contain values for the activity level and the data exchange interactions, i.e., the position in the data flow (biodiversity data user, biodiversity data provider, or both). If multiple members from the same institution indicated different roles in the biodiversity data flow (e.g., one as a data user, another as a provider), their responses were combined to classify the institution accordingly (e.g., as both user and provider). Further details on centrality, activity level, and position in the data flow are provided in the following section.

## Node properties: Centrality, activity level and position in the data flow

We used two functions to calculate network centrality: 1) degree centrality and 2) page rank. Both functions were calculated with the JavaScript library Cytoscape.js [14]. Degree centrality is a measure utilised in network analysis to evaluate the significance of a node within a network. It relies on the concept of counting connections or edges. Put simply, the degree centrality of a node corresponds to the number of edges connected to it. Nodes exhibiting high degree centrality possess numerous connections, indicating their greater centrality or influence within the network. A highly central network member therefore has many connections to other members. Degree centrality is frequently employed to pinpoint key nodes in a network, such as hubs or highly connected individuals in social networks, critical infrastructure nodes in transportation or communication networks, or highly cited papers or authors in citation networks.

Page rank, which is used to evaluate the importance or relevance of web pages on the internet, can be regarded as equivalent to Eigenvector centrality in the context of network analysis. It measures the influence of a node in a connected network. This measure is based not only on the node's direct connections, but also on the centrality of its neighbours. Nodes with higher page rank values are those that are connected to other highly central nodes and vice versa. An impactful network member therefore describes a member that has many connections with highly centralised members.

Activity levels were determined based on the number of EuropaBON project events (n = 17) that a network member participated in. These events included workshops, conferences, webinars, interviews, surveys, social media campaigns, and review processes (please see the EuropaBON website, https://europabon.org, for more details). Members were informed about these events via the

EuropaBON website and social media channels (X, https://x.com/EuropaBon_H2020, and Facebook, https://www.facebook.com/EuropaBONH2020). While scheduling conflicts were not accounted for in the analysis, we recognize that availability constraints may have influenced participation. Additionally, funding was provided to support the involvement of certain members, thereby influencing the level of engagement of those members. Activity levels were assigned as absolute values (0–17) and categorized into three groups: 0=none, 1=low (participation in up to 8 events), and 2=high (participation in more than 8 events). On the dashboard, different colors indicate activity levels: the darker the fill color of each symbol, the more activities the node participated in.

Each node was assigned a position in the data flow (i.e., biodiversity data user, biodiversity data provider, or both) based on the information provided by each member during the registration process (S1 File). On the dashboard, the visualisation uses distinct shapes to represent different member institutions: circles for data users, triangles for data providers, and diamonds for those who serve as both. Additionally, a separate group, "N.A." (Not Available), includes nodes that are not registered members but were identified as data users or providers by members. In the network graph, these non-member nodes are displayed as grey dots.

## Web application architecture

The dashboard has been developed as a fully responsive JavaScript [15] web application using technologies that are licensed as free software using jQuery [16], the Bootstrap 4 framework [17] and various JavaScript libraries, such as Leaflet JS [18], Highcharts [19] and datatables.js [20], a plug-in for the JavaScript library jQuery. The network graph and centrality calculations were generated using Cytoscape.js [14], an open-source JavaScript library for graph visualisation, employing a force-directed FCOSE [21] layout to identify clusters and bridges (S6 File). The API interface between the web application and the server was developed in PHP 8 [22], which runs on the members portal's [13] existing PHP/MariaDB stack using MariaDB v10.6.5 [23] as the database system. The entire website, including the database, is hosted in a VM Docker environment at iDiv (Germany). The deployed website uses Apache web server technology.

The entire source code of the dashboard is available in the EuropaBON GitHub repository (https://github.com/EuropaBON/stakeholder-dashboard), and an archived version has been made publicly available on Zenodo [24]. Further technical specifications are available in the S7 File. For an overview of the web application architecture, see Fig 1. One limitation of the JavaScript library Cytoscape.js was its restricted functionality for advanced statistical analyses. To explore node relationships in more detail, we conducted additional network data analyses outside Cytoscape, which will be presented in the next section.

## Statistical model of network data

This section describes the statistical methods used to examine the factors associated with degree centrality and the number of activities in which EuropaBON members participated. Since both outcomes are count variables (i.e., non-negative integers), we considered a range of count regression models [26,27]. After evaluating several alternatives, we selected the negative binomial regression model as the most appropriate. This model accounts for overdispersion – where the variance exceeds the mean – a common feature in count data that can compromise the reliability of standard Poisson regression by underestimating standard errors and overstating significance levels. The negative binomial model mitigates this issue by incorporating an additional dispersion parameter, offering a more flexible and robust alternative to the standard Poisson model [28]. In the estimated model, the variation in the number of events a member institution participated in, or alternatively, the number of connections between members and non-member participants in EuropaBON (degree centrality), was explained by a set of member-specific characteristics. Using the information provided by individuals in the registration process (S1 File), the following variables were included: position in the data flow, occupational sector, geographical region (distinguishing between countries within Europe that are members of the OECD – the Organisation for Economic Co-operation and Development – and non-OECD countries outside of Europe), realm, and EU directives. The directives variable was based on responses to the question, "Which of these EU directives is of most interest to your work?", which was included in the registration process to better understand the policy contexts informing participants' expertise and focus areas. Although the responses reflect individual-level perspectives rather than institutional mandates, we included

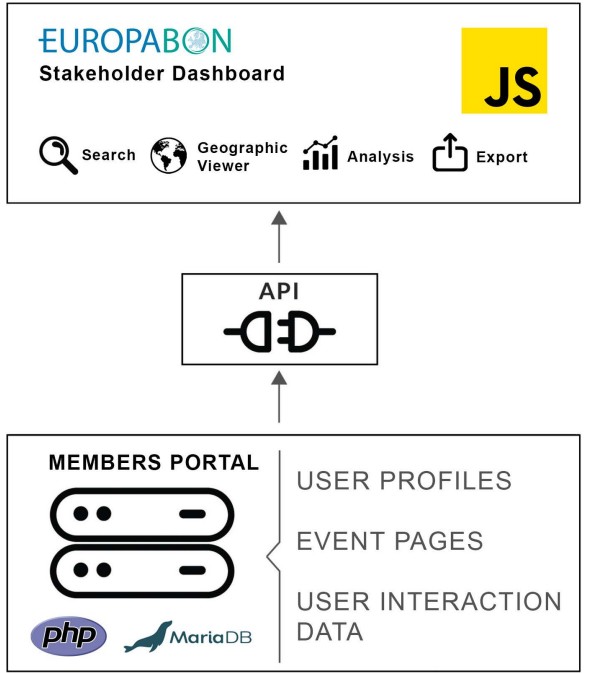

**Fig 1. Web application architecture.** Modified from Velásquez-Tibatá et al. [25], PLOS ONE, Fig 3. Licensed under CC BY 4.0. Source: https://doi.org/10.1371/journal.pone.0214522.g003.

this variable as a proxy for the regulatory environment in which the individual operates. It serves as an indicator of exposure to, or engagement with, EU policy instruments – factors likely to shape the participant's motivation or capacity to engage in EuropaBON activities. While we recognize the potential for reverse causality, we argue that prior exposure to EU directives is more likely to represent a pre-existing condition that influences an individual's engagement, rather than being a direct outcome of their participation. Differentiating between OECD and non-OECD countries outside Europe allowed us to account for potential differences in institutional capacity and access to biodiversity infrastructure, which may affect participation and connectedness within the network. The parameter estimates obtained from the negative binomial regression model were further utilized to compute marginal effects [29], with standard errors calculated using the Delta Method [28]. Marginal effects provide a quantitative interpretation of the model results by estimating the change in the expected value of the dependent variable associated with a one-unit increase in a specific explanatory variable, while holding all other variables constant [30]. This approach is particularly useful for understanding the practical significance of the explanatory variables, as it translates negative binomial regression coefficients into more interpretable units.

## Results

### Dashboard components

All results presented in this study are based on data available as of 22 May 2024. The dynamic and interactive dashboard features multiple components that highlight institutions actively shaping the future European biodiversity monitoring system, along with their connections to key EU projects and infrastructures. The dashboard includes interactive network graphs, charts, and tables (Fig 2 shows a subset of the components). Its main feature is a network graph displaying stakeholders across three dimensions: occupational sector (academia, NGO, governmental organisation, private sector, citizen science, other), realm (terrestrial, freshwater, marine, cross-realm), and EU region (Northern Europe, Western Europe, Eastern

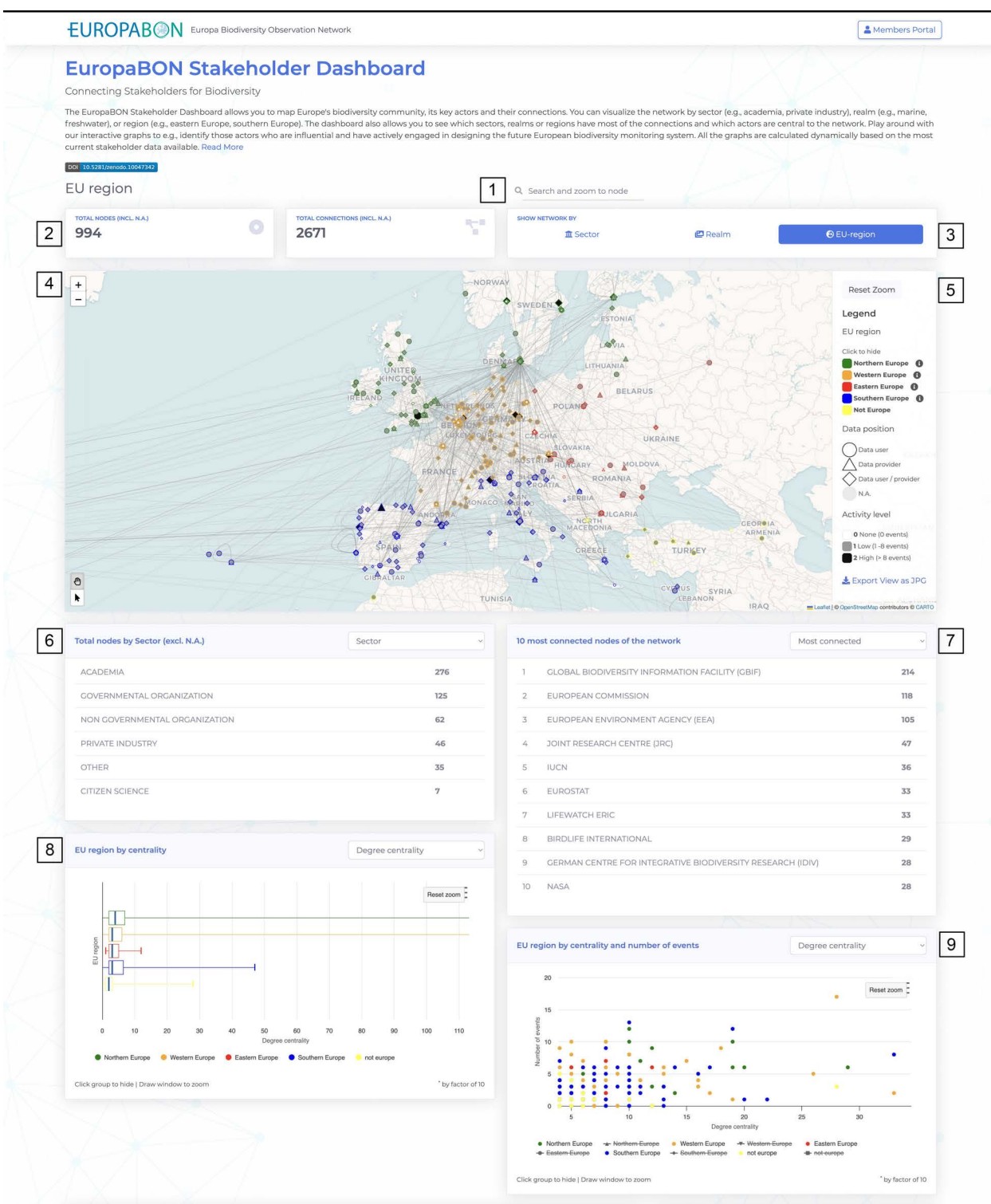

**Fig 2. Subset of the dashboard components.** (1) Search box; (2) Total count of nodes and edges; (3) Show network by dimension (occupational sector, realm, EU region); (4) Network graph; (5) Legend with export option; (6) Total nodes by dimension; (7) 10 most connected/central nodes; (8) Dimension by centrality; (9) Dimension by centrality and number of events. Map tiles in panel (3) are © OpenStreetMap contributors, © CARTO, used under the Creative Commons Attribution 4.0 license (CC BY 4.0).

Europe, Southern Europe, and non-European regions). Graph connections represent data-sharing interactions, indicating whether an institution is a biodiversity data user, data provider, or both. The outlines of the nodes are color-coded to consistently represent these dimensions. Additional dashboard components include: the total count of institutions and connections in the network, the total count of institutions grouped by dimension, the most connected/central institutions, dimension by centrality and number of events, connections to key EU projects and infrastructures, the most connected EU projects and infrastructures, the total count of EU projects and infrastructures grouped by category, and a data table that summarises all data per institution. The dashboard also offers search functionality, geographic mapping of institutions, and data export options. All network graphs, charts, and tables are dynamically updated using the latest EuropaBON member data [13].

## EuropaBON's network

The network currently includes 985 member institutions (i.e., nodes) and 2671 connections (i.e., edges). These are represented by members (n = 590), as well as non-members (n = 395) that have been defined as data providers, -users, or both during the registration process (Table 1). These non-members make up 30% (n = 242) of all of the connections to institutions currently registered in the network.

The majority of member institutions are from the academic sector (n = 278), followed (in order of descending abundance) by government organisations (n = 126), non-governmental organisations (NGOs) (n = 63), representatives of the private sector (n = 46), and citizen scientists (n = 7). While only a small proportion identified their primary affiliation as citizen science, it is possible that more individuals involved in citizen science chose to register under another institutional category, such as academia or NGOs. Almost half (48%) of members indicated that their work spans more than one realm (n = 269), closely followed by members who work in the terrestrial realm (n = 196). Only 12% (n = 66) and 4% (n = 26) of members reported to work on marine and freshwater species/ecosystems, respectively. In terms of geographic biases, Eastern European members are clearly underrepresented in the network, with only 8% (n = 44) of all registrants. This is notably low given that Eastern Europe accounts for approximately 21% of EU member states [31]. Most members and their institutions are located in Western Europe (n = 166), closely followed by Southern (n = 156), and Northern Europe (n = 108). A total of 82 members are from outside Europe. Data users and providers are relatively balanced across sectors; however, NGOs and members from the private industry sector predominantly use data (S2 Fig).

## EuropaBON's stakeholders

The three most connected (i.e., highest centrality) and central (highest page rank) members were the Global Biodiversity Information Facility (GBIF), the European Commission, and the European Environment Agency (EEA) (Fig 3 and Table 2). These three institutions made up > 60% of all connections in the network. When we mapped EU projects and key EU infrastructures that members reported to actively participate in, we found that the majority

**Table 1. List of the top-ten non-member institutions with the largest degree centrality values.**

| Non-member institution | Degree centrality |
|---|---|
| Ocean Biodiversity Information System (OBIS) | 39 |
| EMODnet | 20 |
| Copernicus | 16 |
| Convention on Biological Diversity (CBD) | 15 |
| National Center for Biotechnology Information (NCBI) | 13 |
| National Biodiversity Network (NBN) | 13 |
| GenBank | 11 |
| NOAA Fisheries | 11 |
| European Ocean Biodiversity Information System (EurOBIS) | 11 |
| Barcode of Life Data System (BOLD) | 10 |

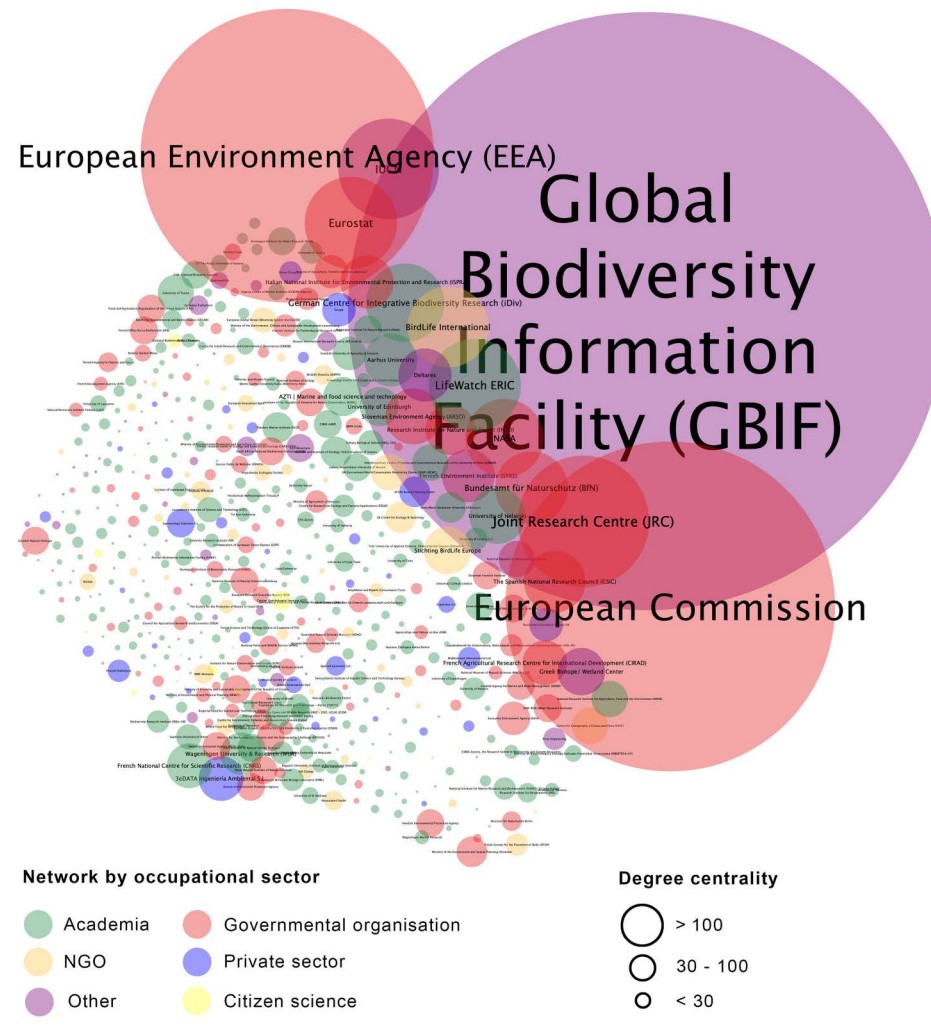

**Network by occupational sector**

- 🟢 Academia
- 🟡 NGO
- 🟣 Other
- 🔴 Governmental organisation
- 🔵 Private sector
- 🟡 Citizen science

**Degree centrality**

- ◯ > 100
- ◯ 30 - 100
- ◦ < 30

**Fig 3. The network visualised by the occupational sector.** The colours represent clusters classified by different occupational sectors, while the size of each node indicates its degree of centrality.

**Table 2. List of the top-ten members with the largest degree centrality and page rank values. Note that degree centrality and page rank values do not display the same rank order for all institutions.**

| Member institution | Degree centrality | Page rank |
|---|---|---|
| Global Biodiversity Information Facility | 214 | 0,121 |
| European Commission | 117 | 0.064 |
| European Environment Agency | 102 | 0.041 |
| Joint Research Centre | 46 | 0.021 |
| Biodiversa+ | 43 | 0.019 |
| International Union for the Conservation of Nature | 36 | 0.012 |
| EUROSTAT | 34 | 0.020 |
| LifeWatch European Research Infrastructure Consortium | 33 | 0.011 |
| Birdlife International | 29 | 0.023 |
| National Aeronautics and Space Administration | 28 | 0.015 |

(34%) were research infrastructures, research networks, or research projects, followed by coordination and/or support networks (19%), and biodiversity tools and technologies (15%). Data repositories (9%), biodiversity observation frameworks (9%), biodiversity monitoring schemes (5%), and intergovernmental organisations/panels (3%), were reported less frequently. The top five most connected EU projects and key infrastructures (in terms of their number of connections) included LifeWatch ERIC (https://lifewatch-eric.openaire.eu), EuropaBON (https://europabon.org), eLTER (https://elter-ri.eu), DiSSCo (https://www.dissco.eu), and Biodiversa+ (https://www.biodiversa.eu) (Fig 4).

**Stakeholder impact and engagement**

In the context of this study, stakeholder impact refers to the degree centrality of the institution with which the registered EuropaBON network member is affiliated. Engagement denotes the number of EuropaBON project events/ activities the member participated in. The calculated marginal effects, along with their corresponding 95% confidence intervals, for the two models – explaining impact and engagement are presented in S3 Fig.

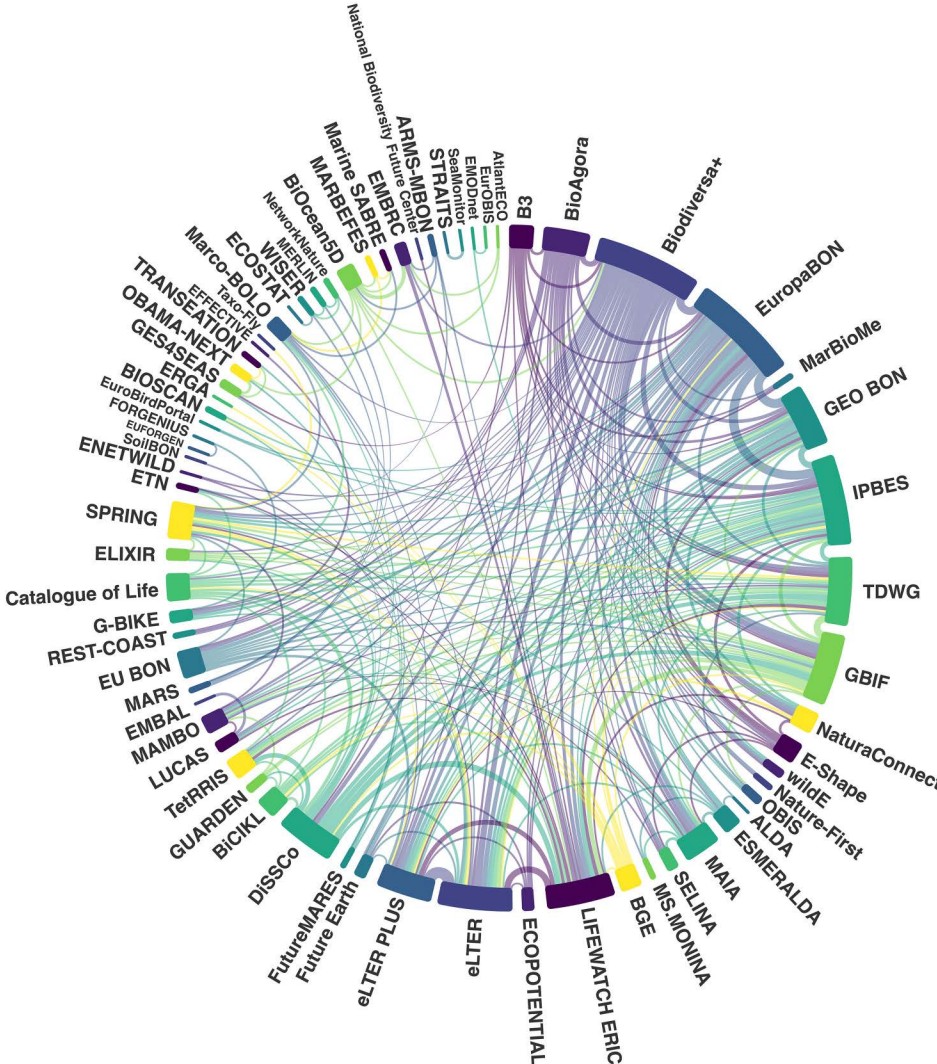

**Fig 4. Connections between EU projects and key EU infrastructures that members reported to be actively involved in/associated with.** The size of the outer bars represents the number of connections each project or infrastructure has with others. The colours facilitate visualisation but do not have any specific meaning.

Members who acted as both data providers and data users, as well as those who solely acted as data providers, had significantly more connections compared to those who solely acted as data users. Specifically, these groups had, on average, 2.4 and 1.3 more connections than data users, with both effects being statistically significant at the 0.05 level. Members from the government sector were significantly more connected than those in academia, while NGOs, the private sector, and citizen scientists showed no difference in connectedness compared to academia. Members not grouped into these sectors had significantly more connections, driven by GBIF, which had the most connections (n = 214). Members in Eastern and Southern Europe, and non-OECD countries outside Europe, were less connected on average than those in Western Europe. Member institutions working across realms were significantly better connected than those focusing on terrestrial species or ecosystems, while there was no difference in connectedness among terrestrial, marine, and freshwater realms.

Additionally, the number of EU directives associated with a member's activities was a good predictor of centrality, with each additional directive corresponding to a 0.16 increase in the number connections. Regarding the number of activities in which members participated in EuropaBON and thus influenced the design of Europe's new biodiversity and ecosystem services monitoring program, those acting as both data user and data provider were involved in significantly more activities than solely data users. Institutions from the academic sector were actively involved in most activities, and the private sector was significantly underrepresented in EuropaBON's stakeholder events. Members from institutions based in Eastern Europe and countries outside Europe (both non-OECD and OECD) participated in significantly fewer events compared to those located in Western Europe. Members whose work focused on the marine realm participated in significantly fewer events compared to those in the terrestrial realm. A positive relationship was observed between the number of EU directives and the number of events in which an organisation participated.

## Discussion

This research highlights the critical importance of fostering an inclusive stakeholder community, both geographically and thematically, for shaping effective European biodiversity policy. While the EuropaBON network demonstrates strong connectivity across regions, sectors, and ecological realms, notable gaps persist, particularly in the participation of Eastern European actors and in the engagement of stakeholders from marine and freshwater ecosystems. At the core of the network, central institutions such as GBIF play a key role in facilitating data exchange and integration, underscoring the value of well-connected hubs within biodiversity data infrastructures [32]. However, rather than merely reaffirming GBIF's centrality, this finding points to the need for greater redundancy and decentralisation, ensuring that no single node becomes a critical point of failure. Strengthening regional hubs or sector-specific platforms could help balance this centralisation and improve the resilience of the broader biodiversity knowledge infrastructure. Additionally, the network's open-access dashboard, which follows the FAIR principles [12], offers significant potential for enhancing collaboration, transparency, and reusability. These findings carry important implications for emerging initiatives, most notably the EU Biodiversity Observation Coordination Centre (EBOCC), which is poised to build on the foundation laid by EuropaBON to advance coordinated biodiversity monitoring across Europe [33].

### Inclusiveness and representation in the network

Our analysis shows that the EuropaBON members network is generally well connected across geographic regions, sectors, and thematic areas. However, there is clear potential to enhance the network's overall resilience by strengthening links with underrepresented stakeholder groups – particularly those in Eastern European countries and the private sector, as well as actors working in marine and freshwater realms. The consistently low levels of participation from Eastern Europe stand out as a key gap that warrants targeted attention. This could be addressed through more focused outreach, such as organising meetings, conferences, and workshops directly within the region to foster local engagement. Likewise, closer interaction with the marine and freshwater communities would enrich the network by creating space for knowledge

exchange on shared challenges, including data standardisation and funding mechanisms. Currently, institutions focusing on terrestrial ecosystems tend to be more active and better connected, highlighting a disparity that limits the network's comprehensiveness. Yet this underrepresentation also presents a strategic opportunity: strengthening connections with marine and freshwater stakeholders could significantly broaden the network's thematic scope. Notably, a large proportion of institutions are classified as "cross-realm," either because data from multiple members within a single organisation were consolidated or because many institutions genuinely engage across multiple ecosystems [34]. This pattern reflects a broader shift toward interdisciplinary approaches in biodiversity research and practice. Enhancing engagement with currently underrepresented realms will therefore not only fill gaps but also align with the evolving nature of biodiversity work – ultimately making the network more balanced, inclusive, and effective.

## Role of key institutions and central actors

Although our findings demonstrate that the EuropaBON network is generally well connected, there is room to improve links with Eastern Europe and increase collaboration with private companies. The network's structure is characterized by strong connections between intergovernmental organisations (e.g., GBIF, EEA, Eurostat, JRC), academic institutions (e.g., universities), and government organisations (e.g., European Commission agencies). Rather than viewing these connections as static features, they should be leveraged strategically, for example, by empowering these central actors to mentor or partner with less connected institutions from underrepresented regions or realms. Among these, GBIF stands out as the most connected node by far, accounting for over 30% of all connections. This highlights GBIF's central role as a global aggregator of biodiversity data and a longstanding hub for data sharing, standardisation, and integration across countries, institutions, and thematic areas. These findings align with those of Bingham et al. [11], who identified GBIF as the most connected actor in both global and European biodiversity informatics landscapes. Institutions in the academic (e.g., NIVA, ISPRA) and government sectors (e.g., EEA, JRC) are the most active in project-related events such as workshops, webinars, surveys, and interviews. This level of activity is not unexpected, as these entities often receive dedicated funding for such engagements. We recommend expanding the range of stakeholders involved in the network and suggest offering increased support for early-career professionals, who may otherwise lack the resources to participate in key meetings and events. Moreover, many of the most influential and well-connected actors, such as BIODIVERSA+, IPBES, eLTER, and LifeWatch ERIC, are themselves cross-realm or serve coordinating roles across multiple ecosystem types. These actors play a crucial role in bridging thematic gaps and highlight the value of integrative infrastructures in network development.

## Implications for biodiversity monitoring and policy

Understanding the structure and composition of the stakeholder network is not an end in itself, but a crucial step toward improving its function and impact. By identifying underrepresented regions, sectors, and realms, targeted actions can be taken to diversify participation, strengthen weak connections, and foster more equitable collaboration. A more inclusive and better-connected network translates into more robust and representative biodiversity data. This is essential for setting conservation priorities, addressing monitoring gaps, and achieving practical outcomes. For example, engaging underrepresented regions such as Eastern Europe helps fill data gaps and better align national monitoring systems with EU-wide goals. Similarly, increased involvement from marine and freshwater experts will improve ecosystem-specific coverage and policy responsiveness. A well-connected stakeholder network also fosters trust, facilitates knowledge exchange, and promotes collaborative ownership of biodiversity initiatives. These factors are important for the long-term sustainability of initiatives like the EU Biodiversity Strategy for 2030 and for responding effectively to biodiversity loss. Looking ahead, the stakeholder network could serve not only as a resource for data mobilisation but also as a governance and decision-support platform, enabling more inclusive co-design and evaluation of biodiversity policy interventions.

## Opportunities and limitations of the dashboard

The interactive dashboard developed in this project provides valuable insights into stakeholder relationships and enables users to identify gaps and opportunities for collaboration. The tool follows the FAIR principles [12] and is both machine-readable and reusable. Our open-source approach allows other networks, such as GEO BON or the Global Youth Biodiversity Network, to use, enhance, and build upon our framework. The dashboard infrastructure includes publicly accessible source code via GitHub, as well as a public API for programmatic access of the data. This allows other systems to directly connect with our network data. To support reuse and adaptation, additional guidance is available upon request, with technical maintenance ensured through 2029. However, ensuring the long-term value of the dashboard requires more than technical openness. Sustained funding and governance structures must be secured to guarantee regular updates, versioning, and quality control. This includes planning for a handover of dashboard maintenance to a permanent host institution, ideally within EBOCC or a related European research infrastructure. Without such planning, the dashboard risks becoming static or obsolete. The dashboard's integration with other European and global platforms also remains limited. For true interoperability, more attention must be paid to aligning metadata standards, API protocols, and visualisation formats with those used by complementary systems such as GBIF, LifeWatch ERIC, and GEO BON. Establishing regular exchanges with these initiatives could help position the dashboard as a central node in the wider biodiversity informatics landscape.

Importantly, the EuropaBON stakeholder network remains open to new members. Individuals and institutions interested in joining are encouraged to get in touch via the official project website (www.europabon.org), where a registration form and contact details are available. New members are regularly integrated into the network and reflected in real time on the dashboard. This ensures that the network remains a dynamic and evolving infrastructure, responsive to new collaborations and expertise. Including this information is essential for demonstrating the long-term relevance, openness, and inclusivity of both the stakeholder network and the dashboard. Clear and sustained communication about how to join is essential not only for inclusivity but also for the operational validity of the dashboard as a living system. However, some technical limitations remain. With a current network size of approximately 1,000 nodes, rendering the graph in the browser takes around 15 seconds. This performance bottleneck is due to the browser-based nature of Cytoscape.js [14]. For larger networks, we therefore recommend conducting calculations server-side. Nevertheless, future improvements in JavaScript, such as enhanced parallelization and GPU support, could further improve performance.

## Future directions and recommendations

The insights gained from this research may inform future initiatives, particularly the EU Biodiversity Observation Coordination Centre (EBOCC), recently launched as a call for tenders by the European Commission Directorate-General for Environment. EBOCC aims to pilot a coordinated biodiversity observation centre based on the EuropaBON proposal, with objectives including harmonised data collection, enhanced collaboration among key actors, support for policy-relevant indicators, and technical guidance to Member States. The stakeholder network and dashboard developed through EuropaBON could play a supporting role in this context, offering a potential foundation for inclusive engagement, cross-sectoral collaboration, and transparent data access. Their integration into EBOCC would help accelerate progress while providing continuity with existing efforts already aligned with policy and user needs. To realise this potential, key next steps include establishing governance structures for the stakeholder network, clarifying dashboard maintenance responsibilities, ensuring interoperability with EU-level data platforms, and defining procedures for onboarding new members and updating the network map.

We also encourage other networks to apply this open, reusable dashboard framework to support broader, more inclusive stakeholder engagement across sectors, regions, and disciplines – an essential step toward more effective biodiversity monitoring in an evolving policy and ecological landscape.

## Supporting information

**S1 File. Full list of questions from the registration form.**
(JSON)

**S2 File. Examples of data cleansing measures for the input "Indicate your position in the biodiversity data flow".**
(DOCX)

**S3 File. SQL Query to group the members according to their institution.** The SQL query aggregates detailed information about the institutions, including their country, events attended, occupational sector, projects, realms, scopes, directives, EU regions, data-sharing interactions, and geographic coordinates.
(SQL)

**S4 File. API endpoints.**
(DOCX)

**S5 File. JSON response of ID 1 as name-value pair.**
(JS)

**S6 File. JavaScript options for the network graph.**
(JS)

**S7 File. Technical specifications and licence.**
(DOCX)

**S1 Fig. Cumulative number of new network registrations over the course of the EuropaBON project.**
(TIF)

**S2 Fig. Distribution of data providers, -users, and organisations that are both, data providers and users, visualised across different occupational sectors.** Citizen scientists are excluded from this figure due to the small number of stakeholders that belong to this occupational category. Stakeholders that could not be categorised into either of these groups were classified as "other" and are also excluded from this figure.
(TIF)

**S3 Fig. Marginal effects of the negative binomial model for degree centrality and number of activities.** Marginal effects and corresponding 95% confidence intervals calculated for explanatory variables grouped into categories (i.e., position in the data flow, occupational sector, geographical region, realm, and EU directives). Dependent variables are stakeholder connectedness (degree centrality) and participation in EuropaBON stakeholder events (number of activities). Filled- and open circles indicate significant (p-value 0.05) and non-significant effects on degree centrality and number of activities that stakeholders participated in. Filled red circles indicate reference levels.
(TIF)

## Acknowledgments

We gratefully acknowledge the expertise and time of our IT colleagues at iDiv, especially Sebastian Eulau and Christopher Zimmermann, in setting up the IT infrastructure and the Docker environment. The authors would also like to thank Emily Wendt (iDiv, Germany) for her support on this project. The authors would like to express their gratitude to the German Centre for Integrative Biodiversity Research (iDiv) for hosting the project coordination.

## Author contributions

**Conceptualization:** Christian Langer, Jessica Junker.

**Data curation:** Christian Langer, Jessica Junker, Ivelina Georgieva.

**Formal analysis:** Christian Langer, Marek Giergiczny.

**Funding acquisition:** Jessica Junker.

**Investigation:** Christian Langer, Jessica Junker.

**Methodology:** Christian Langer, Jessica Junker, Marek Giergiczny.

**Project administration:** Jessica Junker, Henrique Miguel Pereira.

**Resources:** Jessica Junker.

**Software:** Christian Langer.

**Supervision:** Jessica Junker, Henrique Miguel Pereira.

**Validation:** Christian Langer, Ian McCallum, Ivelina Georgieva.

**Visualization:** Christian Langer, Jessica Junker, Marek Giergiczny.

**Writing – original draft:** Christian Langer, Jessica Junker.

**Writing – review & editing:** Christian Langer, Jessica Junker, Ian McCallum.

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
