## [Decision Letter · Decision Letter 0]

22 Jan 2025

PONE-D-24-56281The EuropaBON Stakeholder Dashboard: A dynamic web application to map Europe's biodiversity communityPLOS ONE

Dear Dr. Langer,

Thank you for submitting your manuscript to PLOS ONE. After careful consideration, we feel that it has merit but does not fully meet PLOS ONE’s publication criteria as it currently stands. Therefore, we invite you to submit a revised version of the manuscript that addresses the points raised during the review process.

Please address the comments of the reviewers in detail to increase the clarity and consistency of the manuscript. There are several hints and specific comments on this in the reviews. This also includes providing more context in the discussion.

We look forward to receiving your revised manuscript.

Kind regards,

Florian Borgwardt

Academic Editor

PLOS ONE

Journal Requirements:

2. In your Methods section, please include additional information about your dataset and ensure that you have included a statement specifying whether the collection and analysis method complied with the terms and conditions for the source of the data.

“This is a product of the EuropaBON project funded from the European Union’s Horizon 2020 research and innovation programme under grant agreement No 101003553.”

Reviewers' comments:

Reviewer's Responses to Questions

**Comments to the Author**

1. Is the manuscript technically sound, and do the data support the conclusions?

Reviewer #1: Partly

Reviewer #2: Yes

Reviewer #3: Partly

2. Has the statistical analysis been performed appropriately and rigorously? 

Reviewer #1: I Don't Know

Reviewer #2: Yes

Reviewer #3: Yes

3. Have the authors made all data underlying the findings in their manuscript fully available?

Reviewer #1: Yes

Reviewer #2: Yes

Reviewer #3: No

4. Is the manuscript presented in an intelligible fashion and written in standard English?

Reviewer #1: Yes

Reviewer #2: Yes

Reviewer #3: No

5. Review Comments to the Author

Reviewer #1: The manuscript "The EuropaBON Stakeholder Dashboard: A dynamic web application to map Europe's biodiversity community" provides excellent insight into the EuropaBON stakeholder community, which is essential for building a solid and comprehensive biodiversity monitoring network. The paper focusses on the analysis of stakeholder/member data (entered upon registration) and its visualisation through an interactive, and publicly accessible, online dashboard.

I believe this paper merits publication, but I feel it lacks specific details and clarity in writing when it comes to technical soundness and use of statistics.

One of the main questions pertains to the provenance of the data, which I consider essential for evaluating the representativeness of the results. This consists of two components:

* How have you collected information on the 'edges', i.e. which node contributes data to whom, and receives data from whom?

* How have you dealt with 'translating' information provided by individual members to institution/node level?

Also, please detail any actions you may have taken to assess the representativeness of this edges.

With regards to the statistical analysis, I am not familiar with the negative binomial model used, but -as detailed in the specific comments- miss a clear explanation of the methods and parameters used in this section.

Finally, I presume there’s more to learn from the available data using alternative / additional visualisations. It is no surprise that big (networked) organisations such as GBIF, data ‘requesters such as the EEA and EC and research infrastructures have a high level of centrality. Additionally for some of these nested/networked organisations both the overarching, as well as the regional node organisation feature among the nodes (e.g. OBIS - EurOBIS, [INSDC] - NCBI-Genbank / ENA). It is unclear how this is accounted for. Given the focus on biodiversity monitoring, I wonder if an analysis excluding/hiding those initiatives and having more focus on the institutes who perform actual ‘on the ground’ monitoring would provide additional insights into the European landscape. In it’s current form the visualisation for (1) occupational sectors and (2) realms have only limited information value, and (3) the geographic regions visualisation could possibly be improved by using the colour coding for variables other than geographic region (which is evident from the map).

Specific comments

* The main objectives include identifying skill gaps. I am unsure how this is covered by the data and analysis provided.

* You mention in the introduction "web application that allows users to monitor network changes and activity levels in real time". I would appreciate if you could comment on how realistic this proved to be, what changes are to be expected and which ones would be much harder to pick up (given that I presume users don't update their profile very frequently).

* "Network members provided information essential to this analysis during the network registration process (https://europabon.org/members/register/index)." Note that the full set of questions the user is presented with are only shown upon actual registration, in the interest of this paper it would be useful to include/discuss these in more detail and have the full list included in the supplementary material.

* The phrase "while filtering for verified users with non-empty data provider or data user fields" is unclear. Please elaborate.

* On the nodes API, the sentence "To establish edges, each value in the data user and data provider columns is iterated through, creating connections (from, to) via the nodes' IDs in the edges object." is not entirely clear. Please clarify.

* S3 Figure: this figure seems rather unnecessary.

* "Individual members' information is not disclosed in the dashboard and we grouped stakeholders by their affiliation (i.e., several individual members can belong to the same organisation and thus display as one and the same node)." Please specify how this was done, was this a simple 'distinct' operation?

* The section "Statistical analysis of network data" is very hard to read, also several components of the model lack explicit explanation in the text.

* "and OECD country outside within Europe" - Please check, probably this should be either read "outside" or "within"?

* Table 1 - see earlier comment on nested/distributed organisations

Reviewer #2: General Comments

This paper describes a network analysis identifying key EuropaBON stakeholders, their relationships, and participation patterns, and an associated interactive website that displays the data. The analysis is well executed and I have no suggested changes to that aspect. The website and analysis provide an effective model for other such networks to analyze and display participation patterns (even the larger GEO BON). Importantly, the authors provide full access to the code to reproduce the website and analysis, greatly facilitating such adoption. My general criticism is that the Introduction and Discussion lack detail, including discussion of previous research, and analysis of other similar efforts to characterize conservation networks. That would allow the reader to appreciate how this analysis adds to our current understanding of the functioning of these networks. The Introduction is particularly sparse and contains some vague language about “data” and “communities”, perhaps under the assumption that the reader is already familiar with what types of data are collected in BONs and what types of players are involved. I recommend adding detail throughout, with an eye towards making the story clear to readers with no familiarity with the concept. The Introduction should ideally be expanded into at least 4 paragraphs. Currently, it is lacking in citations and critical background information. The reader must wait until the Results and Discussion to understand why this dashboard is really important and needed. The central paragraphs should provide examples of why it would be useful to visualize this information and why this is currently an unmet need. The Discussion should better highlight how learning about a network can improve it, and how subsequent improvements in the network translate to real conservation outcomes. I provide some suggestions to help assuage these concerns in the Detailed Comments below. Once these are addressed, I feel this paper will make a nice contribution.

Detailed Comments

ABSTRACT

The Abstract is good, but the description of what the dashboard actually displays is vague. Please add a few details in the final 6 sentences of the Abstract that give the reader some idea of specifically what types of data and information are displayed by the dashboard and how it might be used.

INTRODUCTION

1st paragraph, last sentence: “...accessibility of biodiversity data…”: What types of biodiversity data? Please give an example or two.

2nd paragraph, 1st sentence: “...biodiversity communities…”: What is meant by this? Please be more specific.

2nd par, 3rd sentence: “the impact of key contributors”: Again, I find this language to be a bit vague. What kinds of impacts are we talking about? Is this a matter of the amount of data that each country is contributing data to the BON? Please try to be more specific.

2nd paragraph of Introduction: This paragraph (ideally expanded to 2 or 3 paragraphs) should be where most of the background information is provided to the reader. However, it currently has no citations and is generally light on background information. Please expand this to provide the reader adequate background information so that they can fully understand the value of your dashboard contribution.

3rd paragraph of Introduction: “It offers high-level information in one view that can be used to identify occupational sectors, realms, or geographic regions with the most connections and pinpoint the central actors within the network.”:

You haven’t fully established why it is important to visualize the activities of the central actors, nor have you introduced why “occupational sectors” matter

Define what a “realm” is (e.g., how does it differ from a “sector”?).

“The main objective of this study was to 1) map the EuropaBON stakeholder network across sectors, realms and EU regions, identify 2) skills gaps, thematic-, and geographic gaps, 3) data providers and users 4) and key stakeholders, and 5) provide this as a fully responsive, interactive web application that allows users to monitor network changes and activity levels in real time.”: There are logical problems with the flow of language in the above list of objectives including duplication and incorrect placement of verbs relative to list numbers. Rephrase to flow better, perhaps something like: “The main objectives of this study were to map the EuropaBON stakeholder network across sectors, realms and regions, and to identify gaps in skills, thematic program areas, and geography. Our aim was to create a fully responsive, interactive web application that allows users to monitor network changes and activity levels in real time.”

METHODS

Paragraph 1: The number of register EuropaBON members etc. might fit better in the Introduction as background information.

Is “13.09.2024” a correct date format for this journal?

Paragraph 2: “Network members provided information essential to this analysis…” What kinds of information and in what form?

Paragraph 3: “...through right joins in the database, combining information based on common identifiers…” This is unclear and perhaps provides too much detail without actually informing the reader of the end result. The same applies to this “...user details, event statistics, and ISO region information…”. It seems vague, limiting reproducibility, while at the same time providing little insight as to what actually was performed for a general reader. Also, spell out ISO.

“The API has four endpoints:”: Briefly explain here what an API endpoint is and what it does. Without that, the list of API endpoints is less meaningful. The links go to plain text pages which are backend data files that power the dashboard. But without a brief introduction to API endpoints, the reader is left confused about what the function of the links is.

Bullet point 2: “storing them as nodes in JSON format”: Please explain a little what it means to be a node in JSON format. What do they mean in the context of the end user experience?

“Centrality” section: Perhaps add a little about what Centrality would mean in terms of EuropaBON members. The examples you provide about citations and page rank are illustrative but a more concrete example would be nice of what a highly central BON member would be like.

End of Page 7, beginning of page 8: Here is the first time you provide the subcategories in each category such as sector, realm, etc. These should be provided earlier so that it is clear what a realm, etc. is early on.

Beginning of page 8: “...several individual members can belong to the same organisation and thus display as one and the same node…”: I think this information should be made clear earlier on. Until now, it was not clear to me that nodes were organizations and that data from individual members in member portal were being aggregated to inform node characteristics.

Fig. 2 caption: All of these metrics are interesting sounding, but I can’t help but wonder what they will be used for and by who. The background and information regarding utility that is needed to make learning about these metrics interesting should be included in the expanded Introduction.

Page 9, penultimate paragraph of Methods: “...the number of EU directives associated with the stakeholder’s activities…”: I could be wrong, but wouldn’t the number of directives associated with the stakeholder’s activities be (to some extent) caused by the number of activities (the dependent variable) and not vice versa? I recommend carefully considering this possibility.

Table 1: This table would be improved by explaining what centrality means and giving some indication of the range of values expected for this measure (e.g., it is impossible to know if the numbers displayed are high or low, relatively speaking).

RESULTS

Second paragraph of Results: “...are clearly underrepresented in the network, with only 8% (n=44) of all stakeholders…”: This isn’t a supported statement without providing information about what % would be expected if they were proportionally represented. Please either provide this information or else revise the claim.

‘EuropaBON’s stakeholders’ section: “...research infrastructures, -networks, or -projects…”: The hyphens in this sentence are confusing. I suggest spelling out “...research infrastructures, research networks, or research projects…” if that is the intended meaning.

Fig. 4 caption: Please state what the colours represent.

“Factors driving stakeholder…” section, first paragraph: You already state that you used negative binomial models in your Methods, so this is duplication. I would include this information only in the Methods. Much of this paragraph is duplicate information that already occurs (and belongs in) the Methods.

“Factors driving stakeholder…” section, second paragraph: “...these groups had on average by 2.4 and 1.3 more connections…”: I appreciate that you provide effect sizes here. Can you confirm that this means 2.4 and 1.3 more connections on average? Or is it 2.4 and 1.3 times more? Also, more than what, the average? The intercept group?

DISCUSSION

First paragraph: The sentence beginning “Our analysis…” is rather long and would benefit from editing to be more concise or by splitting it into two sentences.

Second paragraph: Can you better explain what the EBOCC is and what it would look like for it to focus on this network?

Third paragraph: You set forth some measures to increase representation, which are good. But how would those translate into better outcomes? That piece is still lacking. Can you spell out why being well connected and inclusive matters?

Fourth paragraph: This paragraph starts out talking about realms, but finishes with a seemingly new topic (a list of key actors). Can you tie these together and provide a concluding sentence to this paragraph?

In general, there is a lack of cited literature in this Discussion and a corresponding lack of connection to previous research and scholarly discussion about conservation networks and planning.

Last paragraph: You rightly point out that other similar collaborative efforts (including GEO BON) would benefit from a similar framework and analysis. I think you could brag a little here about how your solution provides a robust model (and even a complete code) to facilitate such widespread adoption. Well done.

REFERENCES

Again, I think the Introduction and Discussion could use more discussion of previous research and analysis of such conservation networks. That would allow the reader to appreciate how your analysis adds to our current understanding of these networks and how they operate.

Reviewer #3: Dear authors, dear editor,

The manuscript “The EuropaBON Stakeholder Dashboard: A dynamic web application to map Europe's biodiversity community” presents an innovative tool for mapping and analysing biodiversity stakeholder networks in Europe. By identifying key actors, highlighting data gaps, and enabling real-time updates, the dashboard contributes to advancing biodiversity monitoring and stakeholder engagement. While the manuscript aligns well with PLOS ONE's scope, it is not a traditional research article and deviates from the journal's preferred structure. Additionally, the language may be overly technical for a general audience. Although the contribution is valuable, it lacks full originality due to the existence of similar tools in other domains (e.g., biodiversity and environmental networks).

The manuscript contributes to biodiversity monitoring and stakeholder engagement.

The interactive dashboard and open-access code enhance transparency and usability, but the overall work and presentation lacks balance in terms of representation and clarity on how to overcome these issues, e.g. targeted outreach strategies to address geographic or realm underrepresentation. An outlook on the further maintenance is also missing (“at least five years after the project’s life time is a bit vague)

General comments:

A key concern is the lack of clarity regarding how stakeholders became members of the network. Did the authors actively approach key biodiversity actors, or was recruitment primarily passive (e.g., through social media outreach)? The absence of notable stakeholders, such as certain organisations in Switzerland or prominent institutions like the Natural History Museum in London, raises questions about the representativeness of the database. Additionally, incomplete entries (e.g., NHM Vienna) further suggest a lack of systematic inclusion.

Similar concerns arise regarding the measurement of activity levels. How were stakeholders invited to participate in events, and how did the authors account for potential barriers, such as funding constraints or scheduling conflicts? A lack of participation in specific EuropaBON events does not necessarily indicate lower overall engagement in biodiversity-related activities (only because one specific date would not suit them, does not make them less active in the field…). Additionally, the visual use of red to indicate participation below a threshold (e.g., fewer than 8 events) could be misleading, as red is often associated with negative connotations like "alarm" or "failure."

Another major concerns is how the member data are updated; from experience it would need regular reminders to ask members to update their data (including someone sending these reminders and chasing these data); generally I think the requested data do not change too often (except maybe projects organisations are involved in) and they are updated maybe only once a year (maximum), so that it is not really relevant to have this real time component; that of course takes a lot of time when using and filtering the dashboard.

Some of the terminology is not consistent. In the introduction authors talk about “the system”, “the network” and it is not clear what is meant and when it relates to the stakeholder network and/or the biodiversity observation system. This should be checked throughout the manuscript.

Line 64 says “in developing a EuropaBON”: here is a mixture between the European BON (the regional BON of GEO) and EuropaBON (the project); if the same name is used for both, then this should be clarified somewhere; the entire sentence does not make too much sense and needs to be rephrased. What is the “collaboratively designed system”?

In line 60 it is stated that network members were involved in developing the new European biodiversity monitoring system: if this is already in place, then this would need a clear reference! See also 169 onwards.

Methods:

Including a table summarising the information queried during registration would significantly enhance clarity. Additionally, a brief overview of the database structure, including key tables and their relationships, would help readers better understand how data were organized and processed. For instance, the statement “we retrieved data from multiple tables (users, profile, events, country) through right joins in the database, combining information based on common identifiers” is overly technical and could confuse non-specialist readers. Simplifying or illustrating this process would improve accessibility.

The API is described in detail (at least endpoint 2), which might be too specific for the reader and the journal after all. I think the description of the API is not really relevant for the paper and could be moved to the supplementary material. “nodes” and “edges” should be explained to the non-technical reader and the paragraph should be checked for readability and understandability.

For people not familiar with network analysis the term “centrality” and “page rank” need to be introduced before discussing them.

Regarding the position in the dataflow it would be helpful if the data flow could be depicted in a figure.

Regarding the functionality of the dashboard it would be cool if one could search for an organisation in the last table, then click on it and see it in the first map; in that map it would be nice to be able to click on the connections as well and get an information where it links to (this is especially useful if you are zoomed in). Also the number of connections of an organisation could be shown in the mouse-over box.

Results:

Some of the results are clearly connected to the question above how people were asked to join; the underrepresentation of certain realms could be a result of not having actively asked freshwater or marine people. In that aspect, only recording “cross-realm” (without knowing which realms are crossed) also biases the evaluation.

The headline “Factors driving stakeholder centrality and activity” does not correspond with what is really in this paragraph and I would recommend to rename this using a more common word than “centrality”.

Discussion:

The discussion would benefit from a more critical analysis and a broader integration of references beyond EuropaBON-related publications. For example, statements such as “the network would greatly benefit from enhanced interaction with the freshwater and marine communities to exchange on many points of common interest” are vague and lack concrete suggestions for achieving these goals. Similarly, while the importance of ensuring the network's longevity is emphasised, the manuscript does not provide specific strategies or mechanisms for sustaining the network beyond the project’s timeline.

All of my above-mentioned concerns need to be thoroughly addressed in the discussion including recruitment, activity level, maintenance etc.

Additionally, a short discussion on “Stakeholders acting as both data provider and data user, as well as those solely acting as data provider, had significantly more connections compared to those solely acting as data users.“ would be nice as I think this is a valuable result. Also should it be discussed why it is not surprising that GBIF has the most connections.

Further, the authors could strengthen the case for originality by explicitly comparing their tool with existing platforms.

Language wise the paper should be reviewed by a native speaker. Also the use of tenses needs to be re-checked (mixture of past and present tense in the methods for example)

In terms of used citations I would appreciate to have more balanced amount of non EuropaBON references.

Specific comments:

Line 29: rephrase as “one of the most comprehensive networks of biodiversity stakeholders to date was recently developed…”

47: add that there is also a separation in terms of the different realms/biomes

63: the EBV will not be monitored by the system but rather by the stakeholders… maybe “within the system”

64: “in developing a EuropaBON”: here is a mixture between the European BON (the regional BON) and EuropaBON (the project); if the same name is used for both, then this should be clarified somewhere

68-73: these are results and should be skipped here

69: what is meant by “displays of Europe's biodiversity community”?

73 onwards: rethink the punctuation and rephrase as “The main objectives of this study were 1) to map the EuropaBON stakeholder network across sectors, realms and EU regions, 2) to identify skills gaps, thematic and geographic gaps, 3) to identify data providers and users, 4) key stakeholders, as well as 5) to provide this as a fully responsive, interactive web application that allows users to monitor network changes and activity levels in real time.” Or something along these lines

88: add examples which information was queried

93: what are verified users?

95: skip “server’s”

97: I would think the API is not part of the input data and should have a sub-heading on its own

101: which table?

148: please make clear if the web application is the base for the dashboard or if the web application IS the dashboard

154: say that MariaDB actually is a database

157: add “(Germany)” next to iDiv

168: harmonise the use of capital letters for northern, western etc. (it is used with capital letters later on, eg line 198); the hyphen after northern-, western-, eastern- needs to be skipped

186: what does this mean “it can often be modelled”?

193: harmonise “PageRnk” vs “page rank”

198 and everywhere else: harmonise the use of a/no comma after “i.e.” and after “e.g.”

200: harmonise the use of capital letters for the organisations as well throughout the manuscript

202: EU Directives: where does this information come from?

Table 1: spell out BOLD as well; isn’t it “BOLDSYSTEMS” now?

222: remove the space after the “/” and harmonise this throughout

224: skip “currently located in this region”

225: remove the hyphen

236: remove the hyphens

249: remove the hyphen

Figure 4: explain the colours

257: could you give an example for the overdipersion?

259: skip “respectively“

258-261: this is methods and should be moved there

271: rethink the use of “better” rather than “more” as better implies that the quality is higher while it is only the count of connections!

299: to which other networks did you compare this? This should be added.

312: add a second intergovernmental organisation; GBIF to…

314: remove the hyphen

320: remove the hyphen

322: I suppose it should be JRC (Joint Research Centre)

334 onwards: please rephrase this sentence; there are five (!) “and” in there…

338: add ither initiatives that could be relevant

340: replace “these initiatives“ by “this initiative”

344: “synergies” instead of “synergy”

345: I do not understand “recruit non-members” as they will be members as soon as they are recruited, no?

346: consider using an alternative term to “networks” here as generally “network” is mostly used as EuropaBON network in this manuscript (as at the end of this sentence)

6. PLOS authors have the option to publish the peer review history of their article (what does this mean?). If published, this will include your full peer review and any attached files.

Reviewer #1: **Yes: **Aaike De Wever

Reviewer #2: **Yes: **Michael C. Allen

Reviewer #3: No

---

## [Author Response · Author response to Decision Letter 1]

31 Mar 2025

RESPONSE: We have kept to the requirements for the file naming.

2. In your Methods section, please include additional information about your dataset and ensure that you have included a statement specifying whether the collection and analysis method complied with the terms and conditions for the source of the data.

RESPONSE: We have added a statement in our Methods section: This study was conducted in accordance with the EuropaBON Data Privacy and Use Policy, available from the EuropaBON website, which outlines the terms and conditions governing the collection, access, and use of stakeholder data.

“This is a product of the EuropaBON project funded from the European Union’s Horizon 2020 research and innovation programme under grant agreement No 101003553.”

RESPONSE: We have added the information to the role of the funders in the updated cover letter.

All responses regarding the reviewer comments can be found in the uploaded file "Response to Reviewers.docx"

---

## [Decision Letter · Decision Letter 1]

23 May 2025

PONE-D-24-56281R1The EuropaBON Stakeholder Dashboard: A dynamic web application to map Europe's biodiversity communityPLOS ONE

Dear Dr. Langer,

Thank you for submitting your manuscript to PLOS ONE. After careful consideration, we feel that it has merit but does not fully meet PLOS ONE’s publication criteria as it currently stands. Therefore, we invite you to submit a revised version of the manuscript that addresses the points raised during the review process.

We look forward to receiving your revised manuscript.

Kind regards,

Florian Borgwardt

Academic Editor

PLOS ONE

Journal Requirements:

Additional Editor Comments :

Dear authors,

thanks for your revision on the manuscript. I received the feedback from the reviewers and reviewer 3 provides clear guidance how the manuscript should improved. Please consider these comments to streamline your manuscript.

Kindest Regards

Florian Borgwardt

Reviewers' comments:

Reviewer's Responses to Questions

**Comments to the Author**

1. If the authors have adequately addressed your comments raised in a previous round of review and you feel that this manuscript is now acceptable for publication, you may indicate that here to bypass the “Comments to the Author” section, enter your conflict of interest statement in the “Confidential to Editor” section, and submit your "Accept" recommendation.

Reviewer #1: All comments have been addressed

Reviewer #3: (No Response)

2. Is the manuscript technically sound, and do the data support the conclusions?

Reviewer #1: Yes

Reviewer #3: Yes

3. Has the statistical analysis been performed appropriately and rigorously? 

Reviewer #1: Yes

Reviewer #3: Yes

4. Have the authors made all data underlying the findings in their manuscript fully available?

Reviewer #1: Yes

Reviewer #3: Yes

5. Is the manuscript presented in an intelligible fashion and written in standard English?

Reviewer #1: Yes

Reviewer #3: Yes

6. Review Comments to the Author

Reviewer #1: Following the corrections after the first review round, the manuscript “The EuropaBON Stakeholder Dashboard: A dynamic web application to map Europe's biodiversity community” has now greatly improved. The comments were appropriately dealt with, considering the limitations with regard to further development of the online tool. Nevertheless hope I hope that earlier comments pertaining to the functionality, usefulness and up-to-dateness can be tackled during follow-up initiatives of EuropaBON.

Reviewer #3: Thanks to the authors for revising the manuscript and integrating most of my previous comments.

Still, I do have some concerns regarding the publication of the paper.

A lot of time and effort was invested in building up the dashboard and in writing this manuscript. But the sustainability of the dashboard is still very vague and needs to be improved.

As I read it now, the dashboard should be further used for the EBOCC. The paragraph on EBOCC introduces a potentially important initiative but remains vague. It is unclear whether EBOCC is a formal commitment or a proposal under review, and how it relates to the future of the EuropaBON dashboard and stakeholder database. If the intention is to transfer or integrate these components into EBOCC, this should be stated more clearly. Otherwise, the long-term sustainability and purpose of the dashboard remain uncertain.

Also, the manuscript could benefit from explicitly stating whether the EuropaBON stakeholder network is still open to new members and how interested individuals or institutions can join. Since the dashboard is promoted as a real-time tool, clarity on ongoing recruitment and integration of new members is essential for demonstrating its long-term relevance and inclusivity.

The claim that the dashboard is reusable and could be adopted by other networks is valuable but currently vague. The manuscript should specify which components are reusable (e.g., code, database structure, API), what skills are required to adapt them, and whether any documentation or support is available. If establishing “connections with our network” refers to API-level interoperability or shared infrastructure, this should be explicitly described.

In terms language and length, I think some sentences are too lengthy and written too detailed (I even suggested some removements). The methods chapter is far too long. This is a scientific article and not a how-to-manual on building up an API or dashboard. The API endpoints and technical architecture are overly technical for a general audience. SQL queries, API responses, name-value pairs or examples for the attributes can be skipped or moved to Supplementary Material. Some information is just not relevant (“It was developed by the original…”) and should be skipped.

The discussion is still very superficial in terms of on practical implications. It still touches too lightly on critical points like dashboard maintenance, integration with other networks, and tangible next steps (see all my entry points). It repeats results at times (e.g., GBIF's centrality) instead of building on them and the tone is largely promotional.

For better readability, the discussion should be dived into subchapters, e.g.

• Inclusiveness and representation in the Network

• Role of key institutions and central actors

• Implications for biodiversity monitoring and policy

• Opportunities and limitations of the dashboard

• Future directions and recommendations

Still some tense issues are remaining, mixing up use of different tenses in the introduction an methods (e.g. “EuropaBON advances this work by offering a dynamic, continuously updated real-time tool for exploring network structure and engagement, drawing on the FAIR principles” should be in past tense as the project is over.). Also the use of capital letters for attributes or also for the directives needs to be harmonized.

Specific comments

Introduction

2nd paragraph: “… promotes the use of Essential Biodiversity Variables (EBVs)…”: explain what the EBVs can be used for (still not clear to everyone…)

“… have been involved in every step of the development of the new European biodiversity monitoring system…”: this still sounds like the system is already in place; maybe start with explaining what and how far it is (“Although not yet implemented…”) and then add the info how EuropaBON members where involved; please also add the general aim of the new system.

“their participation in project activities”: please change to “EuropaBON project activities”

“Northern Europe, Western Europe, Eastern Europe, Southern Europe, and non-European regions”: this reads a bit weird as there is no Central Europe, but I guess I should have commented on this earlier…

Methods

Input data, 1st paragraph: the very first sentence about the data cleaning is rather lengthy and also the example of only one input field is confusing; you could simply state something like “To ensure consistent institutional representation on the dashboard, we grouped individual member entries by their affiliated institution. When multiple members from the same institution provided different roles in the data flow (e.g., one as a data user, another as a provider), we combined these responses to classify the institution accordingly (e.g., as both user and provider). …” or something along these lines

Data Processing: Network graph:

“Each node is assigned an ID and corresponding attributes (e.g. “id”: “1”, 'label': “German Center for Integrative Biodiversity Research (iDiv)”, "scope": "Global", "group": "datauser", etc.).”: I would skip the example and just list the attributes; please check the use of different types of inverted commas as well as the capital letters in the attributes (see also Terrestrial, Freshwater below).

“The centrality values for each individual node are calculated after the creation of the API and are therefore not included in the endpoint. The appendix S5 file shows an example of the attributes for ID 1.” Skip the first part and move the reference to S5 above to the ID section.

Start a new paragraph at “The API endpoint also contains information about the connections (edges) between the individual nodes.” The sentence is hard to read, maybe rephrase as “This API endpoint also contains information about the connections between the individual nodes, represented by the edges in the network graph.”

Node properties, Centrality, 2nd paragraph: skip the sentence “It can therefore be said that a highly…”

Statistical model of network data: the first sentence says something about the aim of the section (i.e. the analyses); but the aim of the section is to explain the methods! Please skip or change.

“To identify the most suitable approach, we evaluated multiple statistical models …” skip that sentence.

“including non-OECD country outside of Europe, or OECD country within Europe”: I am not clear why this OECD/non-OECD approach was chosen; this should be explained, e.g. “We distinguished between OECD and non-OECD countries outside Europe to account for potential differences in institutional capacity and access to biodiversity infrastructure, which may affect participation and connectedness within the network.”

“The number of EU directives was included as a proxy for the regulatory…”: please indicate the exact purpose of why and how this question was included in the registration process (“Which EU Directives are relevant for your work?”) otherwise this sentence here is a bit unclear as the Directives are not related to the institutions.

Results, EuropaBON’s network, 2nd paragraph: skip “(in order of descending abundance)”.

The sentence “The low number of citizen scientists may…” is confusing, redundant, and speculative and should be changed; suggestion: “Only seven members identified their primary affiliation as citizen science. However, it is possible that more individuals involved in citizen science chose to register under another institutional category, such as academia or NGOs.”

Skip the end of the sentence “suggesting that the region is underrepresented relative to its political and geographic footprint in Europe.”

EuropaBON’s stakeholders, 1st paragraph: add “highest centrality” and “highest page rank”

Figure 3 caption: skip “The visualisation was generated with Cytoscape v3.10.2 [30] using the Compound Spring Embedder (CoSE) layout algorithm [31].” This is Methods.

Figure 4 caption: skip “We selected the "viridis" color scheme…” as this is not relevant. Keep the last sentence though.

Factors influencing stakeholder impact and engagement: change the chapter title to “Stakeholder impact and engagement”

1st paragraph: skip the last sentence “We begin by….”; this is needless.

Discussion

Please see all my comments above.

2nd paragraph: The sentence “Institutes classified as "cross-realm" - those involved in more than one realm - make up the majority of network nodes. This is primarily because, during data processing, information from individual members affiliated with the same institution was consolidated. When members listed different realms for the same institution, these were grouped under the "cross-realm" category. Of course, it is also possible that many institutions today genuinely operate across multiple realms, reflecting the increasingly interdisciplinary nature of research and practice [32].” is confusing and could be improved, e.g. through “The large proportion of ‘cross-realm’ institutions results in part from data processing: when different members of the same institution indicated different realms, their entries were consolidated. However, this may also reflect a genuine trend toward interdisciplinarity, as many institutions increasingly engage across multiple ecosystem types [32].”

7. PLOS authors have the option to publish the peer review history of their article (what does this mean?). If published, this will include your full peer review and any attached files.

Reviewer #1: **Yes: **Aaike De Wever

Reviewer #3: No

---

## [Author Response · Author response to Decision Letter 2]

7 Jul 2025

We have revised the manuscript again and addressed all points raised by reviewer 3. Furthermore, we have checked the reference list again for correctness, as requested by the journal.

---

## [Editor Report · Decision Letter 2]

16 Jul 2025

The EuropaBON Stakeholder Dashboard: A dynamic web application to map Europe's biodiversity community

PONE-D-24-56281R2

Dear Dr. Langer,

We’re pleased to inform you that your manuscript has been judged scientifically suitable for publication and will be formally accepted for publication once it meets all outstanding technical requirements.

Kind regards,

Florian Borgwardt

Academic Editor

PLOS ONE

Additional Editor Comments (optional):

Thanks for your the revisions. I see all comments from the reviewers addressed.
---

## [Editor Report · Acceptance letter]

PONE-D-24-56281R2

PLOS ONE

Dear Dr. Langer,

I'm pleased to inform you that your manuscript has been deemed suitable for publication in PLOS ONE. Congratulations! Your manuscript is now being handed over to our production team.

Kind regards,

on behalf of

Ass.Prof. DI Dr. Florian Borgwardt

Academic Editor

PLOS ONE